# Ultrasoft, Adhesive and Millimeter Scale Epidermis Electronic Sensor for Real-Time Enduringly Monitoring Skin Strain

**DOI:** 10.3390/s19112442

**Published:** 2019-05-29

**Authors:** Jianpeng Zhang, Yuhang Li, Yufeng Xing

**Affiliations:** 1Institute of Solid Mechanics, Beihang University (BUAA), Beijing 100191, China; swat1205@126.com (J.Z.); xingyf@buaa.edu.cn (Y.X.); 2Ningbo Institute of Technology, Beihang University (BUAA), Ningbo 315832, China; 3State Key Laboratory of Strength and Vibration of Mechanical Structures, School of Aerospace Engineering, Xi’an Jiaotong University, Xi’an 710049, China

**Keywords:** epidermal electronic sensors, skin strain, high accuracy, millimeter scale

## Abstract

Epidermal electronic sensors (EESs) possess great advantages in the real-time and enduring monitoring of human vital information compared to the traditional medical device for intimately making contact with human skin. Skin strain is a significant and effective routine to monitor motion, heart rate, wrist pulse, and skin growth in wound healing. In this paper, a novel skin sensor combined with a ternary conductive nanocomposite (Carbon black (CB)/Decamethylcyclopentasiloxane (D5)/Silbione) and a two-stage serpentine connector is designed and fabricated to monitor skin strain. The ultrasoft (~2 kPa) and adhesive properties of the ternary conductive nanocomposite ensure the capacity of the EES to intimately couple with human skin in order to improve accuracy with a relative error of 3.39% at strain 50% as well as a large strain range (0~50%) and gauge factor (GF ~2.5). The millimeter scale EES (~5 mm × 1 mm × 100 μm), based on the micro-nano fabrication technique, consisted of a two-stage serpentine connector and screen print of the ternary conductive nanocomposite. EESs with high comprehensive performance (electrical and mechanical properties) are fabricated to confirm the analytical results and monitor the motion of a human hand. The good agreement between experimental and analytical results paves the way for bettering monitoring of skin growth during wound healing in order to avoid necrosis and scarring. This EES in monitoring the motion of a human exhibit presents a promising application for assisting prosthetic movement.

## 1. Introduction

Epidermal electronic sensors (EESs) have many practical applications [1,2,3,4,5,6] in the real-time and continuous monitoring of human vital information, which demonstrates their huge advantages, such as being ultrasoft, stretchable and ultrathin, over traditional medical sensors. In the past decade, many EESs were developed to monitor human health information, such as blood information (oxygen, glucose, etc.) [7,8,9,10,11], body temperature [12,13], electrocardiogram (ECG) [1], electroencephalogram (EEG) [1], etc. EESs for monitoring skin strains [14,15], including motion [16,17], heart and breath rate [18,19,20], facial expression [21] and wrist pulse [22] are developed to replace wearable sensing devices, for which the features relevant to the comfort of long-term wear are difficult to take into account [23]. The ultrathin (~3 μm) and micro-size (~1 mm) features [24] empower EESs to improve the compactness with skin and the comfort of long-term wear, which encouraged us to develop a novel EESs for the real-time and enduring monitoring of skin strain.

Based on a series of recent breakthroughs in material and technical science, various innovative sensors for skin strain measurement have appeared [25,26,27,28]. Hyper-stretchable and flexible sensors are fabricated to measure large deformation [29,30,31,32]. For example, Wang et al. [29] prepared a commendable skin strain sensor with high fracture elongation up to 2000%. The sensor is composed of multifunctional conductive hydrogels possessing the reversible physical interaction, which imparts the sensor with rapid self-healing ability without any stimuli. Huang et al. [32] used a helix electrohydrodynamic printing technique (HE-Printing) in combination with in-surface self-organized buckling to fabricate a hyper-stretchable self-powered sensor with high stretchability (>300%). However, these sensors not only are perishable and unstable, but also have a low accuracy due to their large size (~5 cm) [29]. Research [33,34,35,36] on conductive nanocomposites composed of conductive fillers, such as carbon black (CB) [35], carbon nanotubes (CNTs) [37,38,39], graphene [20], etc., and flexible polymers demonstrated that sensors fabricated by conductive nanocomposites possess a remarkable durability and stretchability of the polymer and the stable resistance-strain characteristic of the filler. For example, Kong et al. [35] presented a kind of conductive polydimethylsiloxane (PDMS) nanocomposite filled with CB for fabricating sensors with highly linear, good cyclic electrical performance, and mechanical robustness, which can be compared to conventional metal foil gauges. Yin et al. [39] introduced a highly durable ternary conductive nanocomposite, including PDMS, CB and multi-walled carbon nanotubes (MWCNTs), to develop a stretchable strain sensor with remarkable durability (over 105 cycles at 25% strain) and high sensitivity (GF ~12.15). The excellent sensitivity, remarkable durability and large stretchability enable these sensors to detect small vibrations [39] or to map the deformation imaging throughout the large surface for advanced, wearable human-machine interfaces [23,33,35]. 

However, the higher accuracy of real skin strain and the more precise location of the above-threshold region are required to fit other applications, especially to monitor skin strain to avoid necrosis and scarring at the above-threshold region [40]. In order to meet these requirements, the novel sensor should have ultrasoft and ultrathin properties to measure real skin strain and a tiny size to locate the above-threshold region accurately. This important for three reasons. Firstly, the Young’s modulus of pre-existing sensors [35,37,38,39] is comparable to that of human skin. For example, the modulus (>1.8 MPa) [41] of conductive PDMS nanocomposite is larger than the modulus of human skin (0.42–0.85 MPa) [42]. The output strain of these sensors may be less than actual skin strain due to the strengthened effect of these sensors. Secondly, research on mechanical invisibility by Ma et al. [43] and Cai et al. [44] proves that the measurement position located at the upper surface of soft thick polymers, far away from skin, can generate strain isolation. Finally, the average strain of the measuring region is considered as the measurement result of one sensor, which might miss excessive strain due to the effect of strain concentration. A smaller sized sensor can locate the above-threshold region more accurately. The difficulties of connecting external power and preparation process result in large size (~2 cm) of pre-existing sensors [23,35].

In this paper, an ultrasoft biocompatible ternary conductive nanocomposite consisted of CB, Decamethylcyclopentasiloxane (D5) and Silibione is introduced to sense skin strain, and a two-stage serpentine connector is designed to accomplish millimeter scale EESs. The ternary conductive nanocomposite is fabricated as thin-film by screen print and the two-stage serpentine structure ensures the stretchability of the EES. The paper is outlined as follows. The design methods and fabrication process of the EES are described in Section 2. The experiment setup is presented in Section 3. The results and discussion are listed in Section 4. The main conclusions are given in Section 5. 

## 2. Design and Fabrication of the EES for Skin Strain Measurement

### 2.1. Design

The proposed skin sensor consists of two primary components, conductive nanocomposite and a serpentine metallic connector, as shown in Figure 1. Both ends of the serpentine connector, primarily placed on human skin, as shown in Figure 1a, are uncovered in order to intimately make contact with the nanocomposite. The electrical resistance of the connector and other components (~100 Ω), which is far below it of the conductive nanocomposite (>150 kΩ), can be neglected. According to Ohm’s law, the electrical resistance *R* can be written as:(1)$$R=\frac{\rho l}{S}$$
where ρ is the electrical resistivity of the nanocomposite, *l* and *S* are the length and cross-section area, respectively. The volume *V* remains unchanged during deformation, which gives:(2)$$R=\frac{\rho l^{2}}{V}$$

The specific between electrical resistance $R_{E}$ for final deformation and $R_{0}$ for initial deformation can be given as:(3)$$\frac{R_{E}}{R_{0}}=\frac{l_{E}^{2}}{l_{0}^{2}}$$

The strain $\varepsilon =\left(l_{E}-l_{0}\right)/l_{0}$ is substituted into Equation (3), which gives:(4)$$\varepsilon =\sqrt{\frac{R_{E}}{R_{0}}}-1$$

The relation between strain and resistance can be approximately considered as linear by expanding as a Taylor series truncated at the first degree needed under the small deformation [39] but is nonlinear under a large deformation according to Equation (4). Thus, the resistance-strain characteristic of the sensor in this paper is given from Equation (4) to accurately measure skin strain. 

### 2.2. Material

The Silbione silicone (SILBIONE^®^ RT GEL 4717 A&B, Bluestar Silicones Hong Kong Trading Co. Ltd, Hongkong, China) is a soft skin adhesive gel characterized by good adhesion and proven biocompatibility, which is usually used as an adhesive wound dressing and adhesive sheets for scar treatment. The modulus of Silbione (~2 kPa) measured by a dynamic mechanical analyzer (DMA Q800, TA Instruments Inc., New Castle, DE, USA) is far below than that of PDMS (>1.8 MPa) [41], which provides a functional support for neglecting the effects of the modulus to increase measurement accuracy. D5 (Aladdin Industrial Corporation, Shanghai, China) has chemical stability and is widely used in cosmetics and body care products, which determines us to adopt it as a diluter in this ternary conductive nanocomposite. Carbon black (CB) (VXC-72R, Cabot Corporation, Alpharetta, GA, USA) with an average diameter of 30 nm is chosen as the conductive fillers due to its good biocompatibility and stable chemistry.

The two-stage serpentine connector consists of polyimide (PI) and gold (Au). PI (ZKPI-305IIE, Beijing Pome Technology Co., Ltd, Beijing, China) is a universal substrate material with good comprehensive performance, such as heat-resistant 400 °C high electric insulation, low solubility and so on. The chosen conductive material of the connector was gold (Au) since it has excellent chemical (stability) and mechanical properties (ductility), and is biologically harmless, making it desirable among researchers.

### 2.3. Fabrication Process 

The electrical and mechanical properties are characterized by the ternary conductive nanocomposite and the electrical signals are captured by the two-stage serpentine connector. The fabrication processes of the ternary conductive nanocomposite and the serpentine connector are illustrated systematically in Figure 2. The CB is homogenously dispersed into Silbione to synthesize the ternary conductive nanocomposite, as schematically shown in Figure 2a. First, certain weight fractions of CB are fully dissolved in D5 by sonication with water. Then, the A and B components of Silbione are added into the above mixture in turn according to a certain proportion 1:1:2 of A component, B component of Silbione and D5. Finally, the degassing process is applied in a drying oven with optimized temperature 45 °C for 30 min to get uniform and ropy mixture, which is the precursor solution of the ternary conductive nanocomposite. The flexible serpentine connector is fabricated by micro-nano processing, including photolithography, sputtering, lift-off, dry etching and chemical etching, as illustrated systemically in Figure 2b. The films of PI (~5 μm) and photoresist (~1.6 μm) (AZ5214E, Clariant GmbH, Wiesbaden, Germany) are evenly coated by spin-coating with 4000 rpm/min and 3000 rpm/min. The metallic films of Au (~500 nm) and aluminum (Al) (~500 nm) are sputtered by magnetron sputtering. The lift-off process is applied to formulate patterned metallic films, which act as functional components and protective masks for dry etching. Finally, the flexible serpentine connector is peeled off from the glass substrate and cleaned to remove harmful material repeatedly. Figure 2c shows assembling process: The serpentine connector is stuck on human skin. Then, the precursor solution of the ternary conductive nanocomposite (~100 μm) is screen printed over the connector and solidified with 40 °C for 1 h, which is safe for human skin [45]. As shown in Figure 1c, the EES is successfully fabricated with the size of 5 mm × 1 mm × 100 μm for the functional part.

## 3. Experimental Setup

The mechanical behavior of the serpentine connector is critical for the accuracy and repeatability of the EESs due to the fatigue creak. The deformation of the connector can be observed in metallographic microscope (13XF, Beijing Shang Guang Instrument Co. Ltd., Beijing, China) and simulated by finite element analysis (FEA) of ABAQUS. The electrical resistivity of the ternary conductive nanocomposite was measured by four-probe method using a system (RTS-9, 4-Probes Tech., Guangzhou, China) on the basis of the weight fractions of CB. The microstructure images of the nanocomposite containing different CB content were obtained by a scanning electron microscope (EVO/MA15, Carl Zeiss AG., Oberkochen, England) to the different resistivity. The viscosity of the precursor solution of the nanocomposite as a function of the CB content was measured by a digital viscometer (NDJ-9s, Shanghai pingxuan scientific instrument co., LTD, Shanghai, China). The specimens (~100 μm) were prepared by screen printing the precursor solution of the ternary conductive nanocomposite over a glass substrate. The strain-resistance characteristic was given by a measurement system, including a stress equipment (FLR-303, Tianjin Flora Automatic Technology Co. Ltd., Tianjin, China) for quantitative tensile and a digital multimeter (DMM USB-4605, National Instruments, Austin, TX, USA) for measuring electrical resistance. The specimens were located upon a soft substrate made of Ecoflex (Ecoflex-50, Smooth-On, Inc., Macungie, PA, USA), which could maintain at the large deformation (>300%) [41]. Human skin strain was monitored by a system, including personal computer (PC), DMM, printed circuit board (PCB), wires for connecting the DMM to the PC and the PCB, flat cable (FC) for connecting the sensor to the PCB, as shown in Figure 3. 

## 4. Results and Discussion

In terms of mechanical and electrical properties of the EES, the ternary conductive nanocomposite and the serpentine connector were systematically investigated. In order to improve measurement accuracy and range, the ternary conductive nanocomposite with comprehensive performance had to be prepared. The weight percentage of the CB was a key strategy as the D5 was just acted as a diluent emitted during solidifying process. In Figure 4, the effect of the CB content on the electrical resistivity of the ternary conductive nanocomposite showed that increasing the weight percentage of the CB could increase the contact area. The CB was embraced by the Silbione to form a smooth surface at low CB content, as shown in Figure 4a. When increasing the CB content, particles of the CB could encircle the molecule of Silbione to conglomerate, as shown in Figure 4b,c. Micelles density increased with increasing the CB content to reduce the electrical resistivity. Figure 4d shows the log of conductivity as a function of CB content. The black scattered points and red curve represented the experimental and fitting results, which was obtained by a scaling law according to the classical percolation theory [46]. It could be found that the electrical conductivity exponentially increased as the increase of CB content. Besides, the percolation threshold (Pth) of the ternary nanocomposite was 0.47% obtained from the fitting results. The lower electrical resistivity was required to neglect the effect of skin with electrical resistivity (5 − 60 × 10^4^ Ω·cm) [47], but the increase the viscosity of the precursor solution of the ternary conductive nanocomposite, following the increase of the CB content shown in Figure 5, hampered the fabrication of thin film. Thus, 2.5 wt.% of CB content was chosen to fabricate the EESs. 

The modulus of Silbione (~2 kPa) was far below the effective modulus of the connector, which contained PI with 3.17 GPa and Au with 74 GPa [48]. The hyperelasticity and adhesion empowered Sibione to maintain without crack under large deformation. The key factor affecting the application of EES was the deformation of the serpentine connector. Here, finite element analysis (FEA) was performed by using 200804 C3D8R elements for PI and 30000 S4R elements for Au with ABAQUS software to predict the deformation and strains of the connecter. The Poisson’s ratios of the gold and PI were 0.42 and 0.36 [48], respectively. Figure 6 illustrates the deformation of the EES under stretching strains 0%, 25% and 50%, compared with results obtained by FEA. The deformation of FEA could agree well with experimental results and strains in the sensors could be found through the finite element model. When externally stretching strain increases to 50%, the strain of Au can only be 4.05%, which was much lower than destructive strain of gold [49]. These results proved that the sensors can be used to measure a large deformation, which can expand the applications of the EES. 

Considering the above situations, an EES with good comprehensive performance was designed and fabricated. The DMM applied 5 µA to measure the electrical resistance in the measuring process, so the current-voltage (I-V) test [50,51] was carried out by an electrochemical workstation (CHI800D CH Instruments Ins, America). The experimental results showed that the EES exhibited a good linear I-V characteristic, as Figure 7a illustrates. The strain-resistance characteristic was confirmed by experimental results and the GF was obtained from the analytical model, as shown in Figure 7b. The good agreement between analytical results and experimental results exhibited the high accuracy of measurement. For example, the relative error only was 3.39% when the strain of the substrate reached 50%. The high accuracy under large strain of the EES was very important to protect human skin from ischemic necrosis and scar formation during wound-healing monitoring [40]. Figure 8 shows the stability and reproducibility of the EES. Cyclic loading tests were carried out by using a pulse waveform with duty ratio of 50% and frequency of 10Hz. The EES kept good reproducibility and durability over 50000 cycles with the applied strain of 10% and 30%, as shown in Figure 8a,b. However, when increasing the strain up to 50%, the fracture of the serpentine connector of the EES specimen caused a sharp increase in electrical resistance.

Figure 9 showed an application of EESs in motion monitoring of human hand. The EES was placed in a certain part of wrist to monitor the motion of hand. The position of the sensor is shown in Figure 1c where the sensor was parallel to the bone, which is apparent when holding the hand in place. The black line and red line were the actual measurement results and filtering curve, indicating skin strain change with the movement of the hand in Figure 9a. Figure 9b shows skin strain calculated by Equation (4). In this figure, when opening hand, the skin in a certain part of wrist was stretched with a strain of 1.13%. When holding hand, the skin strain was −0.95%.

## 5. Conclusions

In summary, an ultrasoft, adhesive and millimeter-scale skin sensor is developed and fabricated to the real-time and enduring monitoring of human skin strain on the basis of a ternary conductive nanocomposite (CB/D5/Silbione) and a two-stage serpentine connector. The electrical and mechanical properties of the ternary conductive nanocomposite and the two-stage serpentine connector are studied to obtain the EES with a high accuracy, large range and good repeatability. Experimental results show not only the confirmation of analytical results, but also a high accuracy of measurement with a relative error of 3.39% at strain 50%. The high accuracy paves the path for skin-growth monitoring and the strain measurement of a soft structure. In addition, an application of the EES is exhibited in order to monitor the motion of a hand, which presents another potential application for prosthesis movement feedback.

## Figures and Tables

**Figure 1 sensors-19-02442-f001:**
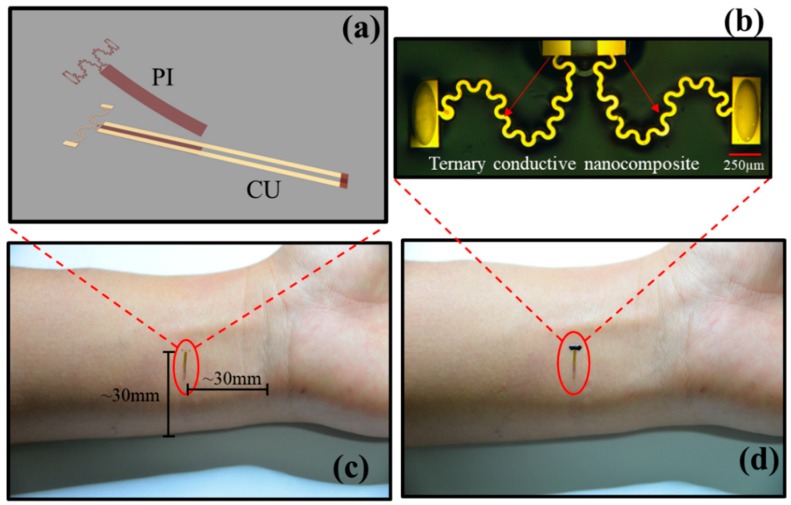
Illustration of the epidermis electronic sensor (EES): (**a**) Schematic illustration of the connector. (**b**) Enlarged image of the EES, the scar bar is 250 µm. (**c**) Image of the two-stage serpentine connector located in human skin. (**d**) Image of the EES.

**Figure 2 sensors-19-02442-f002:**
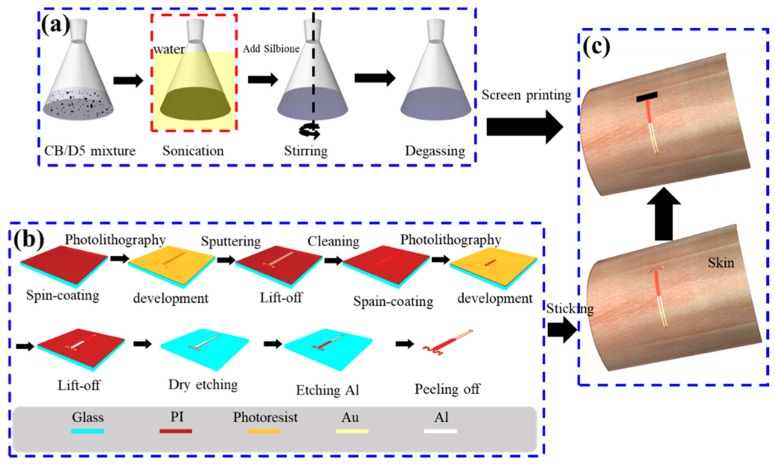
Schematic diagram of the fabrication process of the EES: (**a**) Synthesis process of the ternary conductive nanocomposite. (**b**) Fabrication process of the connector: (**c**) Assembling process of the EES.

**Figure 3 sensors-19-02442-f003:**
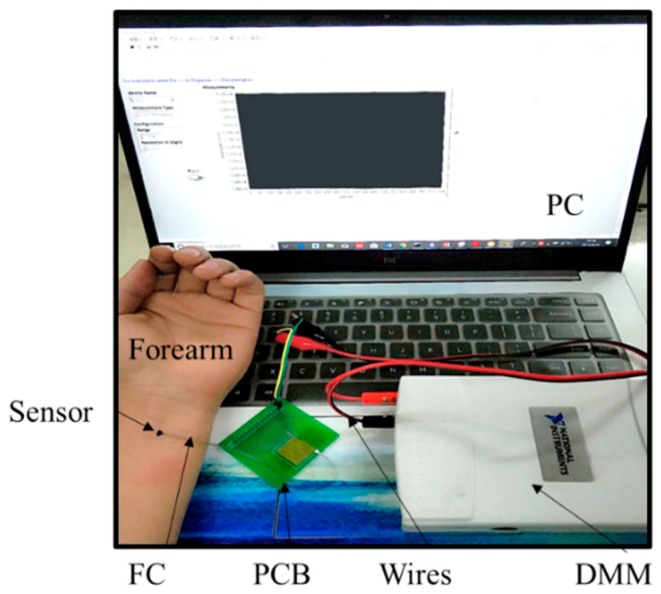
Image of the sensor measurement system.

**Figure 4 sensors-19-02442-f004:**
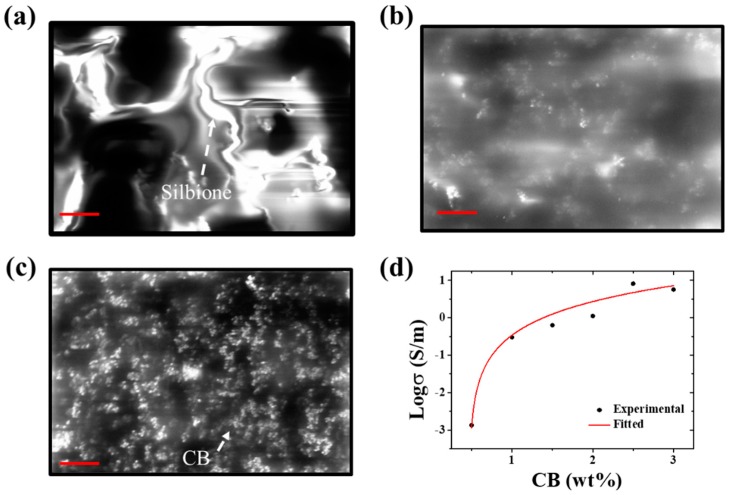
The effect of the CB content on the electrical resistivity of the ternary conductive nanocomposite. SEM image of the ternary conductive nanocomposite containing 0.5wt% CB (**a**), 2wt% CB (**b**) and 3wt% CB (**c**), the scar bar is 8 µm. (**d**) Log of conductivity as a function of CB content.

**Figure 5 sensors-19-02442-f005:**
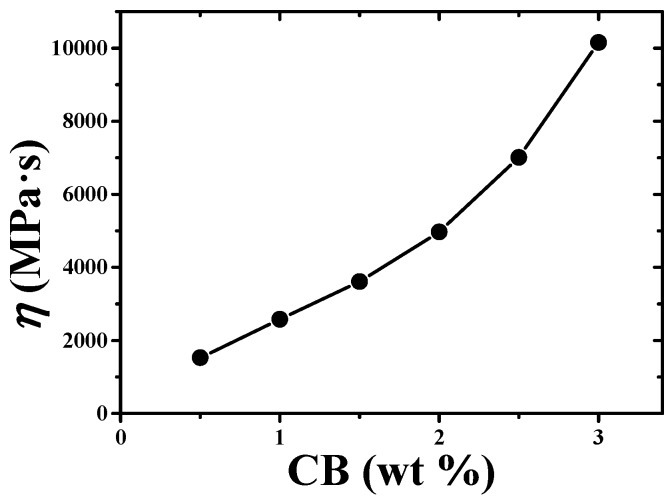
The viscosity of the precursor solution of the ternary conductive nanocomposite as a function of the CB content.

**Figure 6 sensors-19-02442-f006:**
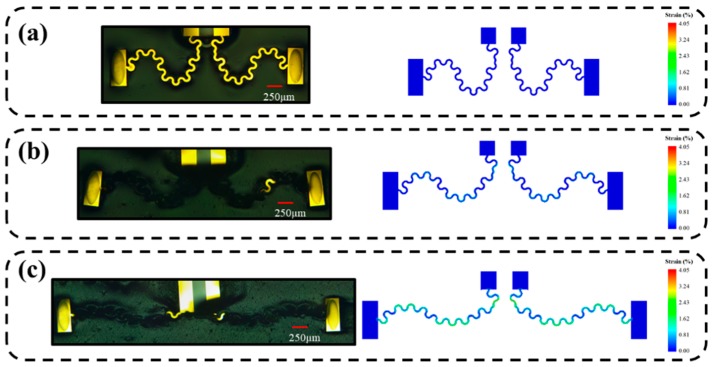
The deformation of the EES under stretching strains 0% (**a**), 25% (**b**) and 50% (**c**), compared with results obtained by FEA.

**Figure 7 sensors-19-02442-f007:**
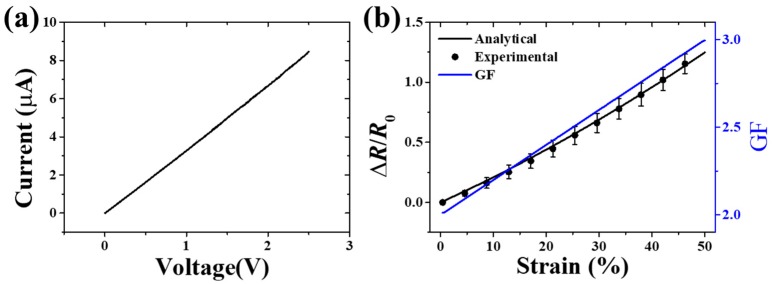
(**a**) I-V test of the EES. (**b**) The results of resistance change ratio and GF under the applied strain of up to 50%.

**Figure 8 sensors-19-02442-f008:**
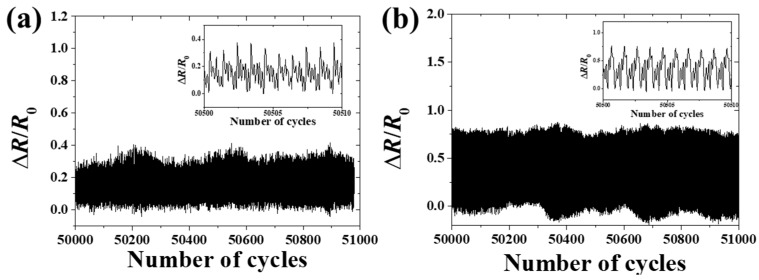
(**a**) The resistance change ratio of the EES under cyclic loading with the strain of 10%. (**b**) The resistance change ratio of the EES under cyclic loading with the strain of 30%.

**Figure 9 sensors-19-02442-f009:**
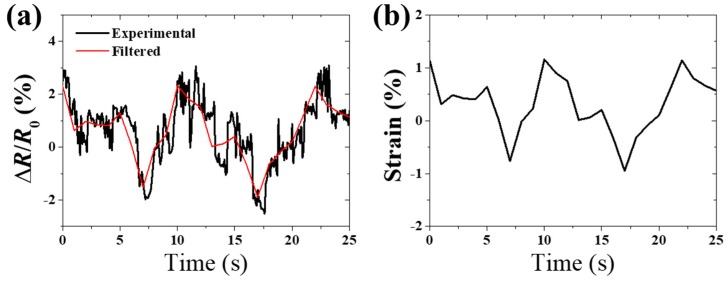
(**a**) The resistance change ratio of the EES under the movement of human hand. (**b**) The human skin strain of a certain part of the wrist measured by the EESs.

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
