# Peer review of "Ultrasoft, Adhesive and Millimeter Scale Epidermis Electronic Sensor for Real-Time Enduringly Monitoring Skin Strain"

_sensors, 2019, doi:10.3390/s19112442_

Round 1
Reviewer 1 Report
In this article, the author demonstrated a novel skin sensor combined with a ternary conductive
nanocomposite (Carbon black (CB)/ Decamethylcyclopentasiloxane (D5)/Silbione) and a two-stage serpentine connector. I think more experiments need to be added.
Following is the detailed list of my concerns, questions and comments:
1. Column 20 “rang” may be “range”.
2. In Figure 1b, for the easy reading, different function parts should be labelled.
3. How to realize the alignment on skin?
4. How to lead wire and realize the signal output?
5. In Figure 3, the electrical resistance of the ternary conductive nanocomposite as a function of the CB content is shown. I think SEM figure should be added to explain the phonemenon.
6. In Figure 5, GF should be calculated.
7. Figure 6 is not sufficient, are those signals of different motion repeatable?
8. The stability and lifetime of devices must be supplied.
9. What physiological signals can the device detect?
Author Response
Point 1:In this article, the author demonstrated a novel skin sensor combined with a ternary conductive nanocomposite (Carbon black (CB)/ Decamethylcyclopentasiloxane (D5)/Silbione) and a two-stage serpentine connector. I think more experiments need to be added.
Response 1: We thank the reviewer’s positive comments. The revised manuscript adds experiments including SEM, I-V test and cyclic loading.
Point 2:Following is the detailed list of my concerns, questions and comments:
1. Column 20 “rang” may be “range”.
2. In Figure 1b, for the easy reading, different function parts should be labelled.
3. How to realize the alignment on skin?
4. How to lead wire and realize the signal output?
5. In Figure 3, the electrical resistance of the ternary conductive nanocomposite as a function of the CB content is shown. I think SEM figure should be added to explain the phonemenon.
6. In Figure 5, GF should be calculated.
7. Figure 6 is not sufficient, are those signals of different motion repeatable?
8. The stability and lifetime of devices must be supplied.
9. What physiological signals can the device detect?
Response 2: We thank the reviewer’s positive comments.
1, This error has been mended in Line 20 and the paper is checked carefully with the help of a professional text editor.
2, The labels of different function parts are added in the new Figure 1b.
3, The position of the sensor is added in the new Figure 1c and the sensor is parallel to the arm.4, The image of the measurement system is added in Figure 3 to illustrate the methods of leading wires and realizing the signal output.
5, The SEM figures of 0.5wt% CB, 2wt% CB, 3wt% CB are added in Figures 4a, 4b and 4c.
6, The gauge factor (GF) is added in Figure 7b.
7, The new experiment on human skin is carried out in thr revised manuscript. and and the repeatability of experimental results are shown in Figure 8.
8, The cyclic loading experiment is carried out to prove the stability, as shown in Figures 7c and 7d. Based the experiment, The lifetime of an EES specimen is 30000 cycles with strain 10%.
9, The sensor is mainly used to detect large deformations of human skin and measure strain of soft materials.
Reviewer 2 Report
“Ultrasoft, adhesive and millimeter scale epidermis electronic sensor for real-tie and enduringly monitoring skin strain” by Zhang et al. describes an epidermal electronic sensor (EES) consisting of a ternary conductive nanocomposite (CB/D5/Silbione) and a two-stage serpentine connector. The sensor possesses a working range of 0~50% strain with 3.39% relative errors. Although the authors used some different materials to fabricate an EES, I could not find any novel aspects in sensor design, performance (sensitivity, working range, cyclic durability, etc.), characterization methodology, integration strategy, and demonstration. I’m seriously skeptical about the author's misunderstanding of “monitoring skin strain”; Almost all types of strain sensors made of soft materials, whether based on composite forms (see many representative works reported by T. Someya) or thin-film forms (see many works regarding metallic nanowires or nanomembranes), can detect and measure the strain in the form of resistance/capacitance variation. The inherently compliant feature of this sensor design can also make it possible to allow for the comfort of long-term wearing, which is verified by a tremendous number of studies from many research groups. From this point of view, it is difficult for me to figure out the author’s claim described in the sentence located in page 1, line 37, which said that “EESs for monitoring skin strains are rarely developed…”.
The geometric design of the serpentine electrodes used in this paper is very common and its electrical characteristic responding to the applied strain was not investigated. The sensitivity (gauge factor) of the sensor is poor and the working range exhibits a modest value (0 to 50%) compared to other studies that used nanowire networks or buckled architectures to achieve >100% available strain, even though the authors claimed in the sentence (page 6, line 238) that “this EES possesses larger range than other stretchable strain sensors”. Additionally, there is no cyclic durability data, which is one of the most important characteristics of strain sensors, and the demonstration of quantifying the wrist skin strain needs further discussion in terms of the definition of strain the authors used. It seems that the wrist skin strain was inversely calculated from the resistance-strain curve of the strain sensor, but it is not that simple to evaluate the exact strain distribution of the wrist skin which has a 2-dimensional strain tensor.
Separately, I strongly suggest that the authors ask a native English speaker or equivalent to assist them with correcting grammar throughout the whole manuscript.
In summary, given the lack of novelty in sensor design and characterization, I do not think this paper meets the requirements needed for publishing in this journal.
Author Response
Point 1: Ultrasoft, adhesive and millimeter scale epidermis electronic sensor for real-tie and enduringly monitoring skin strain” by Zhang et al. describes an epidermal electronic sensor (EES) consisting of a ternary conductive nanocomposite (CB/D5/Silbione) and a two-stage serpentine connector. The sensor possesses a working range of 0~50% strain with 3.39% relative errors. Although the authors used some different materials to fabricate an EES, I could not find any novel aspects in sensor design, performance (sensitivity, working range, cyclic durability, etc.), characterization methodology, integration strategy, and demonstration. I’m seriously skeptical about the author's misunderstanding of “monitoring skin strain”; Almost all types of strain sensors made of soft materials, whether based on composite forms (see many representative works reported by T. Someya) or thin-film forms (see many works regarding metallic nanowires or nanomembranes), can detect and measure the strain in the form of resistance/capacitance variation. The inherently compliant feature of this sensor design can also make it possible to allow for the comfort of long-term wearing, which is verified by a tremendous number of studies from many research groups. From this point of view, it is difficult for me to figure out the author’s claim described in the sentence located in page 1, line 37, which said that “EESs for monitoring skin strains are rarely developed…”.
Response 1: We thank the reviewer’s comments.
1, We agree with the reviewer’s advices and we have mended this sentence. There are many sensors made by conductive composite, but their size is much larger than epidermal electrical sensors with ultrathin (<100µm) and tiny (<25px) properties due to the difficulty of external conductors. Then, we design a serpentine connector to solve this problem.
Point 2: The geometric design of the serpentine electrodes used in this paper is very common and its electrical characteristic responding to the applied strain was not investigated. The sensitivity (gauge factor) of the sensor is poor and the working range exhibits a modest value (0 to 50%) compared to other studies that used nanowire networks or buckled architectures to achieve >100% available strain, even though the authors claimed in the sentence (page 6, line 238) that “this EES possesses larger range than other stretchable strain sensors”. Additionally, there is no cyclic durability data, which is one of the most important characteristics of strain sensors, and the demonstration of quantifying the wrist skin strain needs further discussion in terms of the definition of strain the authors used. It seems that the wrist skin strain was inversely calculated from the resistance-strain curve of the strain sensor, but it is not that simple to evaluate the exact strain distribution of the wrist skin which has a 2-dimensional strain tensor.
Response 2: We agree with the reviewer’s advices. The revised manuscript adds some experiments, including SEM, I-V test and cyclic loading. In Figure 7b, The GF (~2.5) of the EES calculated by the analytical model is close to the GF (~2.2) of the foil strain gauge. The cyclic loading test of the EES with strain 10% is carried out to show the durability and reproducibility in Figures 7c and 7d. The results of the experiment on human skin show the reproducibility in actual application, as shown in Figure 8. However, there are many drawbacks as the reviewer mentioned, such as low GF and difficulty in measuring strain distribution, which also guide us to develop a new strategy in the future.
Point 3: Separately, I strongly suggest that the authors ask a native English speaker or equivalent to assist them with correcting grammar throughout the whole manuscript.
Response 3: We thank the reviewer’s comments. We check grammars of the manuscript carefully with the help of a professional text editor.
In summary, given the lack of novelty in sensor design and characterization, I do not think this paper meets the requirements needed for publishing in this journal.
Reviewer 3 Report
Revisions needed provided in attach (most of English language).

Author Response
Response: We thank the reviewer’s comments. We check grammars of the revised manuscript carefully and mend these errors as the reviewer’s advices
Reviewer 4 Report
Dear authors
I have enjoyed the overall reading of the article. The development of wearable sensors is currently a hot trend, and the minimalistic design of your solution is highly attractive. However, the experimental section of the article needs further improvement, especially the results reported in Figure 6. Additional corrections are next listed.
Line 71
What modulus do you refer in the sentence of Line 71? It may be obvious for you, but not for a non-expert reader. Please clarify.
Line 72
You claimed that the sensor modulus is higher or comparable than that of the skin, and therefore accuracy is reduced. I understand what you mean to say, but probably the word “accuracy” is not the most appropriate. I recommend saying instead “since the sensor Young Modulus is comparable with that of the skin, the resulting measurement is unavoidably affected by the sensor, thus sensor output shouldn’t be trusted”
Line 99
What other sources of resistance are possible? Could you provide further details about this statement?
Section 2.1
This reviewer agrees with the formulation of Section 2.1 (Design), however, polymer composites comprising carbon black usually exhibit a non-linear current-voltage (I-V) characteristic, and therefore, electrical resistance depends upon the sourcing voltage employed in the measuring process [1, 2]. Could you please include an I-V test to determine if your sensor exhibits such non-linear response?
I-V tests are important because it has been demonstrated that sensor’s performance depends upon the operating voltage [2].
Line 213 and Figure 3
Could you please indicate the percolation threshold (Pth) of the ternary composite? This fitting process has been described in multiple studies [3]. Although it is clear you are operating beyond (Pth), it is advisable to explicitly state Pth for comparison purposes.
Figure 6
Four observations are pointed from Figure 3:
· Please include a second y-axis that shows the electrical resistance.
· It seems that different wrist and fist positions yield the same (or very similar) strain; this was observed from the close fist (6% strain) and the open fist (10%). How is it possible?
· Could you please signal intermediate positions in wrist/fist changes? it is hard to determine from the figure what the measuring direction is, i.e. I cannot relate strain changes to wrist/fist position.
· Could you specify the sourcing voltage used for measuring the resistance? As previously stated, the sourcing voltage does play a role in the repeatability of measurements and on sensor’s performance [2, 4].
English corrections
Line 49
The sentence beginning with “which” is not properly connected with the previous sentence, please revise it.
For example, Wang et al. prepared a commendable skin strain sensor composed of multifunctional conductive hydrogels with a high fracture elongation up to 2000%, which the reversible physical interaction imparted with rapid self-healing ability without any stimuli.
Line 65
When enumerating sensor characteristics, it is advisable to replace the comma with “and” after the parenthesis.
Line 66
“Well” is an adverb, and so, “good” should be used in the sentence: “The well mechanical and electrical…”
Line 111
“Deem” should be written in past tense.
Line 163
This sentence must be revised “Then the Al acted as protective masks is removed by chemical etching.”
Line 176
Experiment must be in adjective form as next “Experimental setup”
Sections 3 and 4
In general, experimental setup and results should be written in past tense since they report activities which were already done.
Line 250
This sentence should be revised “Other than other strain sensors, this…”
References
[1] doi: https://doi.org/10.1186/1556-276X-9-369
[2] doi: https://doi.org/10.3390/ma10111334
[3] doi: 10.1039/C4TA03645J
[4] doi: 10.1109/BIOROB.2018.8487226
Author Response
I have enjoyed the overall reading of the article. The development of wearable sensors is currently a hot trend, and the minimalistic design of your solution is highly attractive. However, the experimental section of the article needs further improvement, especially the results reported in Figure 6. Additional corrections are next listed.
Point 1: What modulus do you refer in the sentence of Line 71? It may be obvious for you, but not for a non-expert reader. Please clarify.
Response 1: We thank the reviewer’s positive comments. The references of the modulus are added in the revised manuscript in Line 75
Point 2: You claimed that the sensor modulus is higher or comparable than that of the skin, and therefore accuracy is reduced. I understand what you mean to say, but probably the word “accuracy” is not the most appropriate. I recommend saying instead “since the sensor Young Modulus is comparable with that of the skin, the resulting measurement is unavoidably affected by the sensor, thus sensor output shouldn’t be trusted”
Response 2: We agree with the reviewer’s advices. We revise the sentence as “Firstly, since Young Modulus [35,37-39] of these sensors is comparable with that of human skin, the resulting measurement is unavoidably affected by these sensors. The output strain of these sensors may be less than actual skin strain due to the strengthened effect of these sensors.” In Line 74
Point 3: What other sources of resistance are possible? Could you provide further details about this statement?
Response 3: The other sources of resistance can be neglect and the further details are provided in Line 102 as” The electrical resistance of the connector and other components (~100Ω), which is far below it of the conductive nanocomposite (>150kΩ), can be neglected.”
Point 4: This reviewer agrees with the formulation of Section 2.1 (Design), however, polymer composites comprising carbon black usually exhibit a non-linear current-voltage (I-V) characteristic, and therefore, electrical resistance depends upon the sourcing voltage employed in the measuring process [1, 2]. Could you please include an I-V test to determine if your sensor exhibits such non-linear response?
I-V tests are important because it has been demonstrated that sensor’s performance depends upon the operating voltage [2].
Response 4: We agree with the reviewer’s advices. In Figure 7a, I-V test is carried out and shows that the EES exhibit a good linear I-V characteristic in the measuring process.
Point 5: Could you please indicate the percolation threshold (Pth) of the ternary composite? This fitting process has been described in multiple studies [3]. Although it is clear you are operating beyond (Pth), it is advisable to explicitly state Pth for comparison purposes.
Response 5: We agree with the reviewer’s advices. The percolation threshold (Pth) is 0.47% obtained by fitting process, which refer to the studies [3]. The revised figure and sentence are located in Figure 4b and Line 208
Point 6: Four observations are pointed from Figure 3:
· Please include a second y-axis that shows the electrical resistance.
· It seems that different wrist and fist positions yield the same (or very similar) strain; this was observed from the close fist (6% strain) and the open fist (10%). How is it possible?
· Could you please signal intermediate positions in wrist/fist changes? it is hard to determine from the figure what the measuring direction is, i.e. I cannot relate strain changes to wrist/fist position.
· Could you specify the sourcing voltage used for measuring the resistance? As previously stated, the sourcing voltage does play a role in the repeatability of measurements and on sensor’s performance [2, 4].
Response 6: The new results of the experiment on human skin show the reproducibility in actual application, as shown in Figure 8. The black line and red line are the actual measurement results and filtering curve, indicating skin strain change with the movement of hand in Figure 8a. Figure 8b shows skin strain calculated by Equation 4. In this figure, when opening hand, the skin in a certain part of wrist is stretched with strain of 1.13%. When holding hand, the skin strain is -0.95%.
English corrections
Point 7: The sentence beginning with “which” is not properly connected with the previous sentence, please revise it.
For example, Wang et al. prepared a commendable skin strain sensor composed of multifunctional conductive hydrogels with a high fracture elongation up to 2000%, which the reversible physical interaction imparted with rapid self-healing ability without any stimuli.
Response 7: We revise the sentence in Line 50 as follow.
Wang et al. [29] prepared a commendable skin strain sensor with high fracture elongation up to 2000%. The sensor is composed of multifunctional conductive hydrogels possessing the reversible physical interaction, which impartes the sensor with rapid self-healing ability without any stimuli.
Point 8: When enumerating sensor characteristics, it is advisable to replace the comma with “and” after the parenthesis.
Response 8: We agree with the reviewer’s advices and replace the comma with “and” in line 69 of the revised manuscript.
Point 9: “Well” is an adverb, and so, “good” should be used in the sentence: “The well mechanical and electrical…”
Response 9: We agree with the reviewer’s advices and replace “well” with “good” in line 69 of the revised manuscript.
Point 10: “Deem” should be written in past tense.
Response 10: We agree with the reviewer’s advices and replace “deem” with “considered” in line 116 of the revised manuscript.
Point 11: This sentence must be revised “Then the Al acted as protective masks is removed by chemical etching.”
Response 11: We agree with the reviewer’s advices and delete the sentence to express the preparation process clearly.
Point 12: Experiment must be in adjective form as next “Experimental setup”
Response 12: We agree with the reviewer’s advices and replace “experimental” with “experiment” in Section 3 of the revised manuscript.
Point 13: In general, experimental setup and results should be written in past tense since they report activities which were already done.
Response 13: We agree with the reviewer’s advices. The sections 3 and 4 are be rewritten in past tense.
Point 14: This sentence should be revised “Other than other strain sensors, this…”
Response 14: We agree with the reviewer’s advices and the sentence is revised in Line 250 as follow.
The EES is placed in a certain part of wrist to monitor the motion of hand. The position of the sensor is shown as the Figure 1c and the sensor is parallel to the bone, which is apparent when holding hand.
References
[1] doi: https://doi.org/10.1186/1556-276X-9-369
[2] doi: https://doi.org/10.3390/ma10111334
[3] doi: 10.1039/C4TA03645J
[4] doi: 10.1109/BIOROB.2018.84.87226
Round 2
Reviewer 1 Report
In this article, the author demonstrated a novel skin sensor combined with a ternary conductive
nanocomposite (Carbon black (CB)/ Decamethylcyclopentasiloxane (D5)/Silbione) and a two-stage serpentine connector. I think another experiment need to be added before published.
Authors claimed that the lifetime of an EES specimen is 30000 cycles with strain 10%. However, no figure can be found in the article. Please exhibit it.
Author Response
Point 1:In this article, the author demonstrated a novel skin sensor combined with a ternary conductive nanocomposite (Carbon black (CB)/Decamethylcyclopentasiloxane (D5) /Silbione) and a two-stage serpentine connector. I think another experiment need to be added before published.
Authors claimed that the lifetime of an EES specimen is 30000 cycles with strain 10%. However, no figure can be found in the article. Please exhibit it.
Response 1: We thank the reviewer’s positive comments. The new cyclic loading test is carried out. The experimental results are shown in Figure 8. The revisions in the manuscript are exhibited on line 244-250.” Figure 8 showed the stability and reproducibility of the EES. Cyclic loading tests was carried out by using a pulse waveform with duty ratio of 50% and frequency of 10Hz. The EES kept good reproducibility and durability over 50000 cycles with the applied strain of 10% and 30%, as shown in Figures 8a and 8b. However, when increasing the strain up to 50%, the fracture of the serpentine connector of the EES specimen caused the sharp increase of electrical resistance.”
Reviewer 2 Report
I appreciate the authors’ effort to revise the manuscript.
The thing I pointed out, which was reflected by the author as “Response 1”, is not that easy matter. I strongly recommend the authors to claim their novelty, ultrathin and tiny properties, with reasonable backgrounds in the introduction. For example, why do we need such a tiny-sized conductive composite strain sensor? What is the killer application? In general, strain sensors are used to catch small vibration with excellent sensitivity or to map the deformation imaging throughout the large skin surface for advanced, wearable human-machine interfaces. From this point of view, what is the target feature that the authors are trying to obtain by using a small feature length?
In page 11, line 226, the authors demonstrated 50% stretching of the sensor, but the cyclic loading test was performed using only 10%. What is the deformation or electrical response of the sensor subject to 50% cyclic load?
In summary, it seems to have lots of points that need to be improved.
Author Response
I appreciate the authors’ effort to revise the manuscript.
Point 1: The thing I pointed out, which was reflected by the author as “Response 1”, is not that easy matter. I strongly recommend the authors to claim their novelty, ultrathin and tiny properties, with reasonable backgrounds in the introduction. For example, why do we need such a tiny-sized conductive composite strain sensor? What is the killer application? In general, strain sensors are used to catch small vibration with excellent sensitivity or to map the deformation imaging throughout the large skin surface for advanced, wearable human-machine interfaces. From this point of view, what is the target feature that the authors are trying to obtain by using a small feature length?
Response 1: We agree with the reviewer’s advice. These contents are mended in the new revised manuscript. In section 1, paragraph 3, we describe the novelty, ultrathin and tiny properties of the sensor in detail. The ultrasoft and ultrathin properties are in favour of measuring real skin strain by reducing the strengthened and isolation effects of sensors. Small size sensor can locate the above-threshold region accurately on account of the effect of strain concentration. The above properties are imparted to sensor to fit other applications except vibration detection and deformation imaging, especially to monitor skin strain to avoid it beyond the threshold causing necrosis and scar in medical surgery.
Point 2: In page 11, line 226, the authors demonstrated 50% stretching of the sensor, but the cyclic loading test was performed using only 10%. What is the deformation or electrical response of the sensor subject to 50% cyclic load?
In summary, it seems to have lots of points that need to be improved.
Response 2: We thank the reviewer’s positive comments. The new cyclic loading test is carried out. The new experimental results showed that the EES kept good reproducibility and durability over 50000 cycles with the applied strain of 10% and 30%, as shown in Figures 8a and 8b. The cyclic strain of up to 50% generated the fracture of the serpentine connector of the EES specimen at first loading.
Reviewer 4 Report
The caption of Figure 7 is confussing, I copy it below:
Figure 7. (a) I-V test of the EES. (b) The results of resistance change ratio and GF under the applied strain up to 50%. The resistance change ratio of the EES under cyclic loading with time 0-30s (c) and 10-11s (d).
Bear in mind that sub-figure numbering should be done before figure title (just as in the rest of the manuscript). Hence "(c)" should be placed after 50%, and not after 30s.
Author Response
Point 1: The caption of Figure 7 is confusing, I copy it below:
Figure 7. (a) I-V test of the EES. (b) The results of resistance change ratio and GF under the applied strain up to 50%. The resistance change ratio of the EES under cyclic loading with time 0-30s (c) and 10-11s (d).
Bear in mind that sub-figure numbering should be done before figure title (just as in the rest of the manuscript). Hence "(c)" should be placed after 50%, and not after 30s.
Response 1: We thank the reviewer’s positive comments. The new cyclic loading test is carried out. This error is mended in Figure 8.
Round 3
Reviewer 2 Report
I appreciate the author's efforts to make appropriate revision. I believe the manuscript is now much improved and contains appropriate results to support the work.